# Quantifying the Growth of Preprint Services Hosted by the Center for Open Science

**Tom Narock** [1,*] 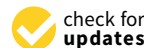 **and Evan B. Goldstein** [2]

1    Center for Data, Mathematical and Computational Sciences, Goucher College, Baltimore, MD 21204, USA
2    Department of Geography, Environment and Sustainability, University of North Carolina at Greensboro, Graham Building, 1009 Spring Garden St., Greensboro, NC 27412, USA; ebgoldst@uncg.edu
*    Correspondence: thomas.narock@goucher.edu

**Abstract:** A wide range of disciplines are building preprint services—web-based systems that enable publishing non peer-reviewed scholarly manuscripts before publication in a peer-reviewed journal. We have quantitatively surveyed nine of the largest English language preprint services offered by the Center for Open Science (COS) and available through an Application Programming Interface. All of the services we investigate also permit the submission of postprints, non-typeset versions of peer-reviewed manuscripts. Data indicates that all services are growing, but with submission rates below more mature services (e.g., bioRxiv). The trend of the preprint-to-postprint ratio for each service indicates that recent growth is a result of more preprint submissions. The nine COS services we investigate host papers that appear in a range of peer-reviewed journals, and many of these publication venues are not listed in the Directory of Open Access Journals. As a result, COS services function as open repositories for peer-reviewed papers that would otherwise be behind a paywall. We further analyze the coauthorship network for each COS service, which indicates that the services have many small connected components, and the largest connected component encompasses only a small percentage of total authors on each service. When comparing the papers submitted to each service, we observe topic overlap measured by keywords self-assigned to each manuscript, indicating that search functionalities would benefit from cutting across the boundaries of a single service. Finally, though annotation capabilities are integrated into all COS services, it is rarely used by readers. Our analysis of these services can be a benchmark for future studies of preprint service growth.

**Keywords:** preprints; postprints; scholarly communication

## 1. Introduction

A preprint is a non peer-reviewed scholarly manuscript that often precedes publication in a peer-reviewed journal. Preprint services have a long history in physics, mathematics, and computer science with ArXiv [1,2], and a less public history in biological sciences dating back to the 1960's [3]. More recently, there has been an explosion of interest in preprints and preprint services [4] and a number of new domain-specific services have emerged. Here, we focus on preprint systems hosted on the Open Science Framework (OSF), a reusable infrastructure developed by the Center for Open Science (COS).

As of 2 April 2019, COS hosts 23 unique preprint services on the OSF platform. These services range from discipline specific (e.g., PaleorXiv for paleontology) to country specific (e.g., INA-Rxiv, which serves the Indonesian research community [4]). The earliest preprint server on OSF began in 2016, with most being created in 2017. OSF preprint services also store other scholarly artifacts, notably postprints (which we define in a similar manner to the criteria of Tennant et al. [5]—non-typeset

versions of peer-reviewed manuscripts), software, and datasets. Authors often submit postprints to COS services to increase access to their research and to archive 'green' open access copies of their work. Many of the COS preprint services were formed through grassroots community action and COS continually reviews applications [6] from communities interested in building a preprint service for a specific discipline, research topic, or geography. A new branded preprint service is created for each community demonstrating need, sufficient organization, and a governance structure. COS supplies a shared toolset for file upload, preprint sharing, persistence, and discovery while the community manages manuscript moderation, governance, and assists in promotion of the service. Despite sharing a common infrastructure, COS preprint services can be configured to match each community's brand, editorial focus, licensing requirements, and taxonomy.

The increased use of preprints has the potential to transform scholarly communication, and many interlinked justifications have been used to spur on the adoption of preprint use (e.g., [7–14]). Preprints allow for rapid public access to new research findings, a way to establish precedent for a research idea, and visibility for early-career researchers. As open access manuscripts, preprints can be read without encountering a paywall, and are visible to the public for commenting, correction, and reviewing even before formal peer review—as such they have the potential to be used in lieu of peer reviewed articles in journal clubs. This is helpful for authors (more reviews allow for more cross fertilization of ideas and error correction) and for pedagogical reasons—training early-career reviewers. Preprints also have the possibility of catalyzing collaborations faster. All of these transformative changes have a high potential of occurring when preprint use and adoption is high within a given field.

There are many metrics that could be used to track the use and adoption of preprints, such as download statistics, preprints citations, surveys and stories regarding the role of preprints, and tracking the use of preprints in applications to grantmaking agencies or promotion and tenure documents. Examples of this work include a recent study of BioRxiv focused on the >30,000 preprints that were submitted in the first five years of the service [15] and the summary stats displayed by PrePubMed [16] for many preprint services in the biological sciences.

Here we focus on quantifying the growth of nine OSF-hosted preprint services using submission metrics, preprint/postprint ratios, topic analysis, coauthorship network analysis, and annotation. Narock et al. [17] previously examined the first year growth in EarthArXiv, the OSF hosted preprint service for the Earth Sciences. We expand on this previous work and investigate multiple OSF-hosted services spanning a range of disciplines. We leverage the growth in preprinting and preprint services hosted by a single entity to examine questions and understand commonalities in service growth. These questions allow for assessment of preprint services that go beyond analysis of single services. Specifically we investigate how many documents are stored on each service, their breakdown (preprints vs. postprints), if they are eventually published in OA venues, coauthorship networks, if these services are seeing commenting/annotation, and the overlap in topics amongst the OSF services. Our goal in this work is to investigate different preprint services to explore commonalities and trends across disciplines and communities.

## 2. Methods

The Center for Open Science (COS) currently hosts twenty three preprint systems across a variety of domains. In order to obtain meaningful statistics, we limited our study to systems that meet four criteria. First, a service must have at least 100 total manuscripts. Second, we look exclusively at services with English language manuscripts. Third, the manuscripts must be accessible through the OSF Application Programming Interface (API) [18] which will enable us to analyze all of these services through programmatic means. Fourth, the service must have a domain/community focus as our intent is to compare preprint usage across multiple communities. General purpose systems—e.g., OSF Preprints and Thesis Commons—were excluded. Overall, nine preprint services meet these four criteria. Data from these nine services were collected via the API in December of 2018.

The software used in this study [19] and the data that we obtained from the API [20] are freely and openly available.

Throughout this study we reuse open science definitions presented in [21]. Specifically,

- Gold Open Access—articles published by a journal website that are freely and openly available. These journals are included in the Directory of Open Access Journals (DOAJ)
- Green Open Access—articles self-archived by an author in a free repository, but published in a pay-for-access journal. For the context of this article, the free open access repository is a COS systems.
- Hybrid Open Access—articles where the author has paid an article processing charge to make their specific work freely available and open access, but the article is published in a pay-for-access journal.

Our analysis requires classifying manuscripts as preprint or postprint. Determining whether a paper deposited in a COS service is a preprint or postprint is done by first extracting all articles returned by the OSF API containing a peer reviewed DOI. This peer-reviewed DOI is fed to the Unpaywall API [22]. Unpaywall is an open-source browser extension and API used for identifying open access versions of a publication that are behind a paywall [21]. We use the Unpaywall API to determine the date of journal publication for each peer-reviewed article. If the COS publication date precedes the journal publication date, then we classify the COS manuscript a preprint—it was published to a COS system prior to being published by a journal. If the COS publication date occurs after the journal publication date, the manuscript is classified a postprint—it was deposited in a COS service after journal publication. Errors reported by the Unpaywall API are reported here as unclassified. Specifically, these are cases where the Unpaywall API did not recognize valid DOIs. Any COS manuscript not having a peer-reviewed DOI is classified as a preprint.

Preprint and postprint classifications are based off of data available via the OSF API. In other words, we are dependent on authors either (1) voluntarily returning to their COS system and updating their records with the peer-reviewed DOI (for a preprint) or (2) providing the peer-reviewed DOI for their postprint submission. Some authors may not comply with this protocol—either intentionally or unintentionally—for a variety of reasons (e.g., they may forget, or the final publication venues may not mint a DOI). For this reason our results likely underestimate the total number of COS manuscripts that appear in peer-reviewed journals.

In order to estimate the extent to which we underestimate the number of COS manuscripts appearing in peer-reviewed journals, we chose 12 random preprints from each system—excluding LawArXiv—for a total of 96 papers. We excluded LawArXiv as this domain often published in venues not assigning DOIs. The 96 preprint titles were manually entered into Google Scholar to identify cases where peer-reviewed versions of the paper existed, but the authors did not update the corresponding COS record. We found 35 such cases (36%). While our 36% false-positive rate is less than what was found in bioRxiv [15]; we caution that our sample size was smaller and that our approach looked for exact title matches only. Our approach will have missed cases where the manuscript title was changed during peer review. Thus, the COS false-positive rate for publication appears to be on par with other well established preprint systems.

Of the 35 false-positives, 25 were preprints—the manuscript was submitted to a COS system before it appeared in a journal. In these 25 cases, the authors failed to return to their COS system to add the peer reviewed DOI. Ten cases were postprints—the authors uploaded an already peer reviewed article to a COS system without supplying the peer reviewed DOI.

We also use the Unpaywall API to investigate where peer-reviewed COS manuscripts are published, and if these venues are open access (OA). The Unpaywall API returns information on which journal articles are published as Gold Open Access Articles, in venues that are included in the Directory of Open Access Journals (DOAJ) [23]. Note that individual papers published in journals that are not included in the DOAJ may still be OA (i.e., via Hybrid Open Access). As a result, this analysis

underestimates the number of COS manuscripts that are published as OA peer-reviewed journal articles. All data from the Unpaywall API was obtained on 4 December 2018.

We use the Disciplines hierarchy for topic analysis. Disciplines is a three-tiered taxonomy of academic subject areas and is the result of a collaboration between bepress and the University of California's California Digital Library [24]. All COS preprint systems use the Disciplines hierarchy, and authors of manuscripts submitted to COS repositories self-tag their work for enhanced discoverability. The Disciplines hierarchy can be further extended by COS communities to better reflect their domain, but communities are restricted to only adding more specific terms. Thus, all COS systems share a top-level topic hierarchy. We use this hierarchy to examine topic overlap across the different COS preprint services. The OSF API returns author assigned Discipline tags for each COS manuscript. We then apply the Szymkiewicz-Simpson overlap coefficient [25] to account for disparity in number of tags. The Szymkiewicz-Simpson coefficient is a similarity measure, related to the Jaccard index, that measures overlap between two sets (*X*, *Y*). The coefficient accounts for differing set sizes by measuring how much the smaller set is a subset of the larger. Formally, the coefficient is defined as:

$$overlap(X, Y) = \frac{|X \cap Y|}{min(|X|, |Y|)} \tag{1}$$

The overlap coefficient ranges from 0 (no overlap) to 1 (smaller set is found in its entirety in the larger set).

Finally, all of the COS services use Hypothes.is [26] for commenting and annotation. To investigate the extent of commenting and annotation on the selected COS services, we searched for the number of annotations made on each service using the search function on the Hypothes.is site [27]. The search was performed on 4 April 2019.

## 3. Results

### 3.1. Basic Metrics

Basic statistics from each of the nine preprint services meeting our requirements are presented in Table 1. Four services were founded in 2016 (LawArXiv, EngrXiv, PsyArXiv, and SocArXiv), while the remaining five were founded in 2017.

**Table 1.** Paper and Author counts for 9 COS preprint services

| Preprint System | Domain | Total Papers | Total Authors | Distinct Authors |
|---|---|---|---|---|
| PsyArXiv | Psychology | 3534 | 12,439 | 7342 |
| SocArXiv | Sociology | 3034 | 5337 | 3017 |
| LawArXiv | Law | 905 | 1186 | 463 |
| EarthArXiv | Earth and Planetary Science | 567 | 2353 | 1659 |
| EngrXiv | Engineering | 362 | 961 | 664 |
| MarXiv | Marine Science | 324 | 960 | 627 |
| LISSA | Library and Information Science | 139 | 257 | 175 |
| MindRxiv | Mind and Contemplative Practices | 121 | 413 | 285 |
| PaleorXiv | Paleontology | 112 | 343 | 236 |

Beyond the total paper count for each service, we can visualize the growth of each service through plots of the cumulative manuscripts in each repository (Figure 1). Qualitatively, each of the 9 services are growing in linear or superlinear manner.

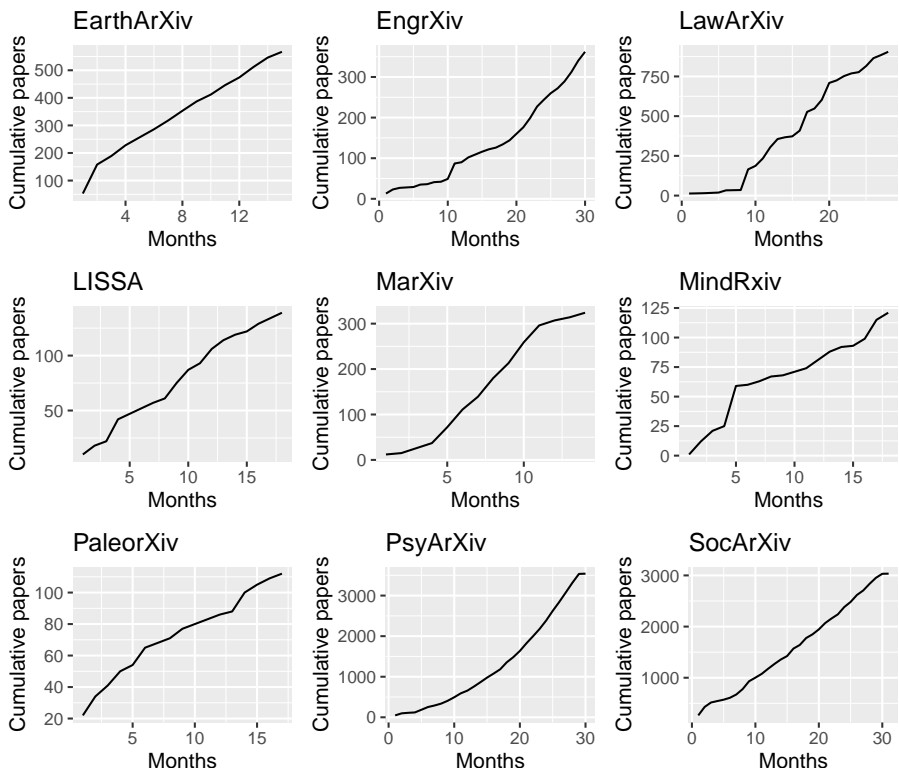

**Figure 1.** Cumulative paper submissions for the 9 services, data is recorded monthly. Note that values on both the X axis (Cumulative paper submission count) and Y axis (months since founding) are different for each panel. Each service will vary based on discipline specific paper production rates.

Data obtained from the OSF API enables us to determine the number of authors that have contributed manuscripts to each service. Total author numbers are shown in Table 1. We filter the total author list for duplicates to find how many distinct authors have contributed to each service. This Distinct Authors number (Table 1) is a proxy for the breadth of a service. A low percentage of distinct authors is indicative of a small group of researchers repeatedly submitting to the service. Conversely, a high percentage of distinct authors indicates more uptake of the service. Distinct authors account for 39–71% of submissions across the 9 COS services. LawArXiv has the lowest distinct author percentage (39%), and EarthArXiv has the highest (71%). The other services cluster between 57–69% distinct authors.

*3.2. Preprints vs. Postprints*

Both preprints and postprints are accepted at the nine COS services allowing us to calculate the total number of submissions of preprints vs. postprints for each COS service. Calculating the ratio of preprint submissions to postprint submissions can tell us how COS services are being used by their respective communities. A larger number of submitted postprints (small ratio) is indicative of a system being used to host already peer-reviewed research. In such a scenario, users may not be interested in the benefits of preprints (e.g., rapid publication, feedback on initial research), or there might be discipline norms that disincentivize preprint adoption. In these cases, authors use COS services to host already published research. Conversely, a larger number of submitted preprints (large ratio) is indicative of researchers using COS services for pre-peer-review benefits (e.g., citability, priority, feedback).

Preprint submission totals, postprint submission totals, and the ratio between these two manuscript types are shown in Table 2 for the nine services we investigate. We believe the high ratio for LawArxiv is because many documents published there do not report a DOI, and thus even

peer-reviewed documents in LawArXiv do not get classified as 'Postprints' using our methodology. We therefore exclude LawArXiv from subsequent analysis of preprint/postprint ratio. Preprint to Postprint ratios range from 7.5 (PsyArXiv) to 0.4 (PaleorXiv).

**Table 2.** Preprint and Postprint count for 9 COS services.

| Preprint System | Papers with Peer Reviewed DOI | Confirmed Preprints | Confirmed Postprints | Unclassified | Total Preprint to Postprint Ratio |
|---|---|---|---|---|---|
| PsyArXiv | 652 | 3111 | 413 | 16 | 7.5 |
| SocArXiv | 900 | 2230 | 768 | 36 | 2.9 |
| LawArXiv | 41 | 867 | 27 | 11 | 32.1 |
| EarthArXiv | 262 | 376 | 187 | 4 | 2.0 |
| EngrXiv | 91 | 285 | 69 | 8 | 4.1 |
| MarXiv | 183 | 190 | 134 | 0 | 1.4 |
| LISSA | 32 | 111 | 21 | 7 | 5.3 |
| MindRxiv | 38 | 87 | 33 | 1 | 2.6 |
| PaleorXiv | 84 | 31 | 81 | 0 | 0.4 |

These preprint to postprint ratios are not fixed through time. Narock et al. (2019) discussed how the initial growth of EarthArXiv was driven by a frontloading of the service with postprints for the launch of the service. As a result it took time for preprints to start outpacing postprint uploads on EarthArXiv. We investigate the temporal evolution of these ratios in Figure 2. First we examine the evolution of the cumulative ratio (Figure 2a). Three services have trends of increasingly being used as preprint servers (PsyArXiv, EarthArXiv, LISSA). Slightly downward trends are seen for the remaining five services. Submission ratios are also examined yearly, which might be useful if submission trends are variable (i.e., frontloading a service with postprints). Again, three services show trends of being used more for preprints—PsyArXiv, EarthArXiv, LISSA—and the growth in preprint usage is more pronounced (Figure 2b). We want to remind readers that postprint designation requires the peer-reviewed DOI to be entered by the author in the COS service, so these results may overestimate the number of preprints in each service.

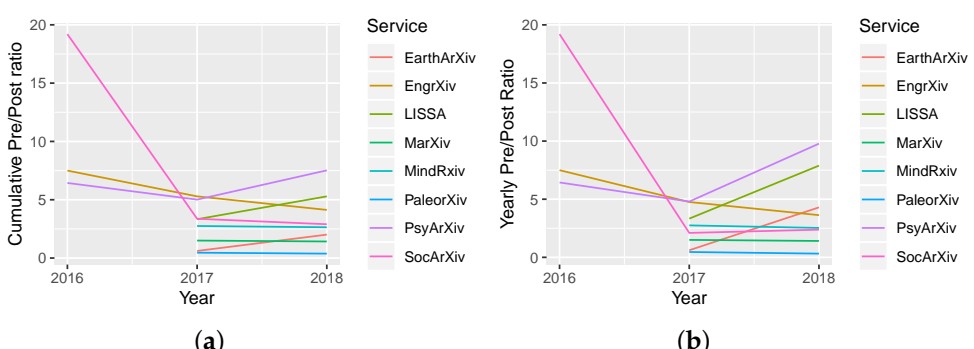

**Figure 2.** Temporal evolution of ratio between preprint submission to postprint submissions. (**a**) cumulative ratio (**b**) single year ratio.

### 3.3. Open Access

Using the Unpaywall API allows us to assess where COS manuscripts are published—both the number of unique journals and if the journal is included within the DOAJ. We do this analysis for preprints that are eventually published in peer-reviewed venues (Table 3) and for postprints (Table 4). The number of preprints that are eventually published in Gold Open Access venues varies widely across COS communities, ranging from 0% (MarXiv, LISSA, and MindRxiv) to 33% (PaleoarXiv). There are a number of postprints published Gold Open Access before being deposited in their respective Green Open Access COS systems (Table 4). For example, 39% of MindRxiv, 35% of SocArXiv, and 33% of LISSA postprints were published in Gold Open Access journals (Table 4).

**Table 3.** Journal and OA status for COS Preprints.

| COS System | Preprints | DOAJ Preprints | Preprints Not DOAJ | Preprint DOAJ % | # of Preprint Unique Journals |
|---|---|---|---|---|---|
| PsyArXiv | 226 | 29 | 197 | 13% | 128 |
| SocArXiv | 96 | 16 | 80 | 17% | 86 |
| LawArXiv | 3 | 0 | 3 | 0% | 3 |
| EarthArXiv | 71 | 9 | 62 | 13% | 47 |
| EngrXiv | 14 | 3 | 11 | 21% | 12 |
| MarXiv | 49 | 0 | 49 | 0% | 13 |
| LISSA | 4 | 0 | 4 | 0% | 3 |
| MindRxiv | 4 | 0 | 4 | 0% | 4 |
| PaleorXiv | 3 | 1 | 2 | 33% | 3 |

**Table 4.** Journal and OA status for COS Postprints.

| COS System | Postprints | DOAJ Postprints | Postprints Not DOAJ | Postprint DOAJ % | # of Postprint Unique Journals |
|---|---|---|---|---|---|
| PsyArXiv | 413 | 32 | 381 | 8% | 242 |
| SocArXiv | 768 | 265 | 503 | 35% | 335 |
| LawArXiv | 27 | 1 | 26 | 4% | 23 |
| EarthArXiv | 187 | 8 | 179 | 4% | 84 |
| EngrXiv | 70 | 4 | 66 | 6% | 55 |
| MarXiv | 134 | 13 | 121 | 10% | 64 |
| LISSA | 21 | 7 | 14 | 33% | 18 |
| MindRxiv | 33 | 13 | 20 | 39% | 23 |
| PaleorXiv | 81 | 8 | 73 | 10% | 40 |

*3.4. Coauthorship Network*

We examine the co-authorship network in each COS system by building an undirected network. If A co-authored a manuscript with B and C, then A, B, and C become nodes in a network with bidirectional links between them (e.g., A–B, A–C, B–C). We do not apply any weighting to the edges—If authors A and B co-authored multiple manuscripts together this adds no new information to the graph. Furthermore, there is no temporal component for our analysis—the nodes and edges in our co-authorship graphs are all connections over the years 2016 through 2018.

The COS infrastructure requires submitting authors to have an account prior to publishing a manuscript. This has the benefit that subsequent submissions by that author will reuse the account details and the author's name will appear exactly as it did on prior submissions. Co-authors, however, do not require an account—although, co-authors can be added to the COS database at the submitting author's discretion. It may also be the case that a submitting author has multiple COS accounts. This results in name ambiguity—such as nicknames or initials being used, e.g., Matthew on one paper and Matt on another—and leads to uncertainty in the network analysis.

We looked at all pairwise combinations of the 24,249 author names appearing in our data using the following methodology

1. Compare last names. If they are the same, then continue to the next step; otherwise, this pair has no conflict
2. Compare the first names. If they are an exact match, then there is no conflict as this is an author submitting another paper. If the first letter of the first name is a match (but the full first name is not an exact match), then mark this pair for manual inspection
3. Manually disambiguate the pairs identified in the above step.

Following this methodology we identified 63 potential author duplicates. We were unable to conclusively disambiguate these authors. They remained in the data during our analysis and we note this introduces uncertainty into our results. Yet, we feel the small number of potential duplicates implies our results are a good approximation of the true COS networks. Further, this implies that

submitting authors are actively adding their co-authors to the COS database for unique identification and reuse in other papers.

Each COS system has its own network, with a number of connected components (subnetworks within the entire network). An example is shown in Figure 3, where the network is composed of three connected components. Although not shown here, an isolated node not connected to any other nodes in the network is also considered a connected component. For each of the individual COS system networks, we examine the number of connected components (Table 5) and their size distribution (i.e., the number of nodes in each component; Figure 4). For clarity, the largest connected component is removed from each panel and Figure 4 shows the distribution of the remaining components. PsyArXiv has the greatest percentage of authors in the largest connected component (55%) with LawArXiv having the fewest (3%). All nine COS systems show many small connected components, and fewer large connected components.

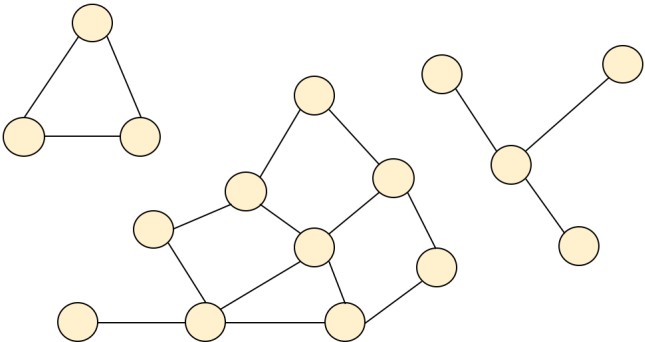

**Figure 3.** An example of 3 connected components within an idealized network.

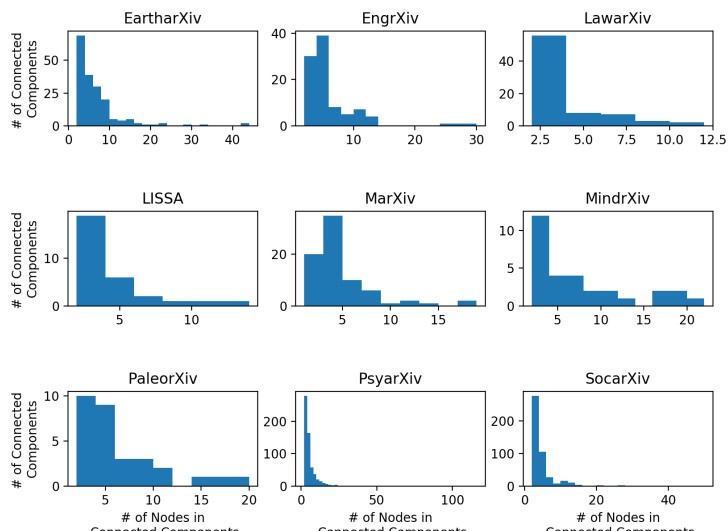

**Figure 4.** Distribution of connected components (excluding the largest) amongst the nine COS systems. Note that both X and Y axes are different for each panel, but the bin width (along the X axis) is fixed at 2 nodes.

**Table 5.** Coauthorship network statistics for 9 COS services.

| Preprint System | Distinct Authors | Authors in Largest Connected Component |
|---|---|---|
| PsyArXiv | 7342 | 4009 |
| SocArXiv | 3017 | 428 |
| LawArXiv | 463 | 16 |
| EarthArXiv | 1659 | 491 |
| EngrXiv | 664 | 32 |
| MarXiv | 627 | 240 |
| LISSA | 175 | 15 |
| MindRxiv | 285 | 54 |
| PaleorXiv | 236 | 43 |

*3.5. Topic Overlap*

We visualize topic model overlap by comparing the total number of keywords in common between any two of the services using the Szymkiewicz-Simpson overlap coefficient (Figure 5). This topic overlap analysis reveals several interesting examples: LawArXiv overlaps with EarthArXiv on the topic of Public Health while EarthArXiv, SocArXiv, and MarXiv all contain manuscripts tagged with Environmental Studies. Similarly, SocArXiv and MarXiv both have manuscripts addressing Public Affairs, Public Policy, and Public Administration.

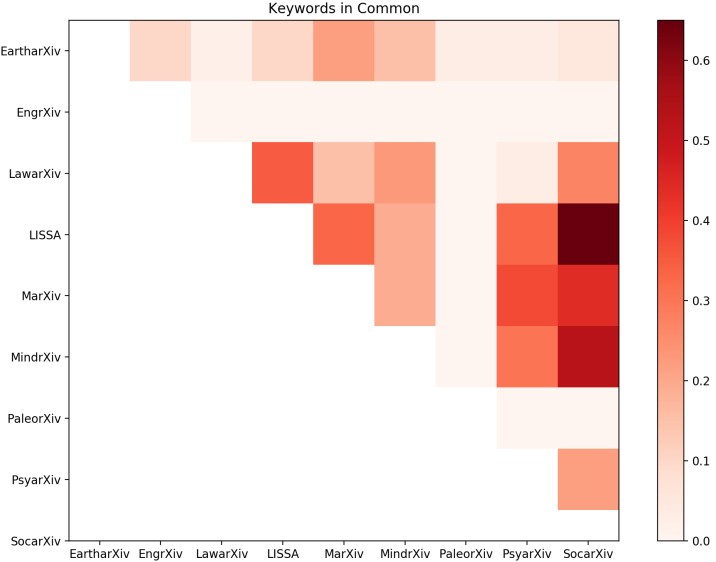

**Figure 5.** The Szymkiewicz-Simpson overlap coefficient for each pairwise comparison with 0 indicating no overlap and 1 indicating the smaller set of keywords is a subset of the larger set.

*3.6. Annotation*

A total of 135 Hypothes.is annotations were found across the >9000 papers on the nine services (Table 6). The number of annotations in total, and across each service is low. A notable distinction between services emerges when the number of annotations are normalized by the total number of papers at a service. PaleorXiv has nearly five times more annotations per manuscript as the next closest service PsyArXiv.

**Table 6.** Annotation counts for 9 COS services.

| Preprint System | Hypothes.is Annotations | Annotations per Manuscript |
|---|---|---|
| PsyArXiv | 89 | 0.03 |
| SocArXiv | 14 | 0.01 |
| LawArXiv | 0 | 0.00 |
| EarthArXiv | 12 | 0.02 |
| EngrXiv | 2 | 0.01 |
| MarXiv | 3 | 0.01 |
| LISSA | 0 | 0.00 |
| MindRxiv | 1 | 0.01 |
| PaleorXiv | 14 | 0.13 |

## 4. Discussion

### 4.1. Basic Metrics

The COS metrics can be placed in context by comparison with a recent analysis of all 37,648 preprints uploaded to the the biological preprint system bioRxiv.org by Abdill and Blekhman [15]. bioRxiv's recent high of 2100 papers per month is equivalent to the total number of preprints in the two biggest services we investigated (Psyarxiv and Socarxiv; Table 2). It is important to remember that bioRxiv was founded in 2013, has many more years of community outreach, and encompasses many productive subdisciplines. The growth rates for all COS Services (Figure 1) have trends of increasing submissions and adoption by each community. A key path toward continual adoption is an increase in community outreach efforts and normalizing the practice of preprinting in each discipline. The preprint-to-postprint ratio for 8 of the 9 COS services investigated here indicates these systems are becoming preprint repositories (Figure 2); although, there is considerable uncertainty around the classification of preprints—an avenue for future COS community engagement.

Open Access for peer-reviewed publications can occur through a variety of mechanisms (e.g., [21]). By using the Unpaywall API we have determined the number of peer reviewed articles from the COS services that appear in Open Access journals listed in the Directory of Open Access Journals (DOAJ). Inclusion in the DOAJ is typically used to denote Gold Open Access Journals (e.g. [21,28,29])—where a journal makes articles open for reuse and directly accessible free of charge on its website. The percentage of articles published in Gold Open Access varies by discipline, but Piwowar et al. [21,30] reports 5.5% of Earth and Space, 4% of Engineering and Technology articles, 4.7% of Psychology articles, and 1.2% for Social Science articles. The percentage of COS preprint articles that appear in Gold OA venues range from 0% (LawArXiv, MarXiv, LISSA, and MindRxiv) to 33% (PaleorXiv). The percentage of preprint articles in DOAJ venues from the 9 COS systems we study are low, but, where present, exceed the values presented by Piwowar et al. [21,30]. What we find surprising is the high percentage of COS postprints that appear in DOAJ publications (Table 4). It is unclear why, having already published Gold OA, authors would also publish in the Green OA COS systems.

An additional place where links to Green OA versions of peer-reviewed articles could be useful is in Wikipedia, one of the largest websites in terms of global web traffic [31], an entry point to scholarly literature [32], and a site with page views that eclipse those of the primary literature (e.g., [33]). Only 29% of the references in Wikipedia refer to free open access papers, and an additional 10% have a free open access version available but without an active link to the OA version [34,35]. We examined the March 2018 release of Wikipedia references [36] to look for COS preprint DOIs in the Wikipedia data and found only 3. In theory, Wikipedia should be finding COS preprints and updating its citations automatically via the OABot [37]. Yet, OABot requires repositories to expose their content in a specific manner and automated identification of open access replacement citations is difficult due to changes made during peer review (i.e., change in title). The updating of Wikipedia references with available

OA versions has become a manual process as a result [38]. The searching and linking of postprints from COS services with Wikipedia is a potentially fruitful direction of work.

### 4.2. Coauthorship Network

Diversity of research topics within any COS preprint system likely guarantees there will always be fragmentation within the network. Furthermore, disciplines where single author contributions are more common will always have more fragmentation. We see initial evidence of this as the COS systems with the smallest percentage of all authors in the largest connected component are the services where author-to-paper ratio is near one (i.e., many single author papers). Thus, we expect the number of connected components in any COS system will always be greater than 1. We expect that the largest connected component will encompass more authors (e.g., [39]) as COS services continue to grow, authors discover new collaborators, and more papers are stored on the service. The percentage of authors in the largest connected component for each service ranges from 3–55%, much lower than the numbers reported in analysis of larger paper collections (e.g., [39,40]) and large academic conferences [41]. Our expectation is that the largest connected components for each COS service will grow toward a value that mirrors these previously published results.

Connected components are also informative when the primary research topic of each component is analyzed. For example, EarthArXiv has far more Geology papers (136) than Soil Science papers (10). Quantifying the size and primary research area of connected components can be a beneficial long term strategy for preprint systems. This type of analysis can help inform a governance team where they should focus resources to maximize inclusion and idea sharing. Network analysis, in general, and connected component analysis, in particular, can help prioritize engagement and outreach strategies and assist in diversifying ideas and community.

### 4.3. Topic Overlap

COS preprint systems are unique in that they share both a common infrastructure and also a common high-level topic keyword hierarchy. This feature allows for increased discoverability of related work by enabling integrated search across multiple domains. Our analysis shows that seemingly disparate domains contain topic overlap. Overlap in the topics present in multiple preprint services suggests that, for specific services and specific topics, authors could be routed to the more general preprint search offered by COS [42]. Our analysis points to two potential next steps for the COS community as they develop a Joint Roadmap for Open Science Tools (JROST) [43]. First, search can be performed across several preprint services. For example, when users search for certain topic keywords within a preprint service, the search results can be returned from other preprint services. This is currently not the default and requires manually visiting the aforementioned generalized COS search page and resubmitting the search query. Our data show there is enough topic overlap that this could be the default. For example, when searching for articles on the topic of Public Health in EarthArXiv, users might also be presented with results from LawArXiv, which also has many topic overlaps. This method of search—based on a shared taxonomy and infrastructure—might enable COS services to accelerate cross-fertilization, helping researchers identify and develop transdisciplinary or convergent approaches [44], and catalyze collaborations.

A second next step for the preprint community is to develop keyword alignment with non-COS preprint systems (e.g., bioRxiv, arXiv, and PeerJ Preprints) to enable broader search. This overlaps with existing efforts to specifically align domain taxonomies as well as more general efforts in the computer science community focused on computational logic and machine reasoning. Examples of the former include the Research Data Alliance and their Data Foundation and Terminology Working Group [45], which may serve as an ideal neutral space for such work. Examples of the latter include the International Semantic Web Conference's annual workshop on ontology alignment [46], existing science-based ontology repositories [47], and specific community-based groups such as the

Earth Science Information Partners' Semantic Technologies Committee [48]. Leveraging this existing research can enhance preprint engagement.

### 4.4. Annotation

For COS systems, we find only 135 Hypothes.is annotations across the >9000 papers of the nine services we examined. It remains to be seen if COS and other scholarly users will embrace annotation as a means of providing comments or commentary on papers [49]. This low level of annotation mirrors the slow adoption of moderated comments on bioRxiv—a 2016 study reported that only 10% of preprints have comments [50]. Aside from COS services introducing the ability to annotate, other scholarly publishing venues have embraced the trend: The American Geophysical Union introduced Hypothes.is integration into the peer review process [51], and eLife has had a partnership with Hypothes.is since 2016 [52].

### 4.5. Final Remarks

Our analysis of the nine largest English language COS preprint services available through the OSF API is intended as a way of quantifying preprint growth. Submission numbers, preprint-to-postprint ratios, coauthorship network statistics, and annotation counts give readers and service users an understanding of how many people are using the service, and in what capacity. There is current work focused on permitting download statistics to be retrieved via the OSF API. Future work incorporating download statistics will provide another dimension of usage. Finally, topic overlap measured by keyword appearance suggests that many services are complementary to one another. We suggest future preprint service features that focus on refining search, or providing other means to sort and find relevant information.

**Author Contributions:** Conceptualization, T.N. and E.B.G.; methodology, T.N. and E.B.G.; software, T.N. and E.B.G.; formal analysis, T.N. and E.B.G.; writing—original draft preparation, T.N. and E.B.G.; writing—review and editing, T.N. and E.B.G.; visualization, T.N. and E.B.G.

**Funding:** This research received no external funding.

**Acknowledgments:** We thank COS and specifically the EarthArXiv community for stimulating discussions on preprints and preprint services.

**Conflicts of Interest:** The authors declare no conflict of interest.

## Abbreviations

The following abbreviations are used in this manuscript:

| | |
|---|---|
| COS | Center for Open Science |
| OSF | Open Science Framework |
| DOI | Digital Object Identifier |
| OA | Open Access |
| DOAJ | Directory of open access journals |
| API | Application Programming Interface |

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
