# Peer review of "Quantifying the Growth of Preprint Services Hosted by the Center for Open Science"

_publications, doi:10.3390/publications7020044_

Round 1

Reviewer 1 Report

Narock and Goldstein present a census of nine preprint servers hosted by the Center for Open Science, presenting metrics about the number of papers, authors and comments on each service and and evaluating patterns in publication (particularly in open-access journals) and co-authorship.

This paper is a useful analysis of a topic (preprints and OA) that is frequently discussed but much less frequently studied, which makes foundational analyses like this one valuable. The authors make sensible decisions at every step of the process: which services to evaluate, which metrics to focus on, and how to present this data. Figure 1 is a good example of their approach—it is concise but effective. Their introduction and discussion sections are well-supported with a broad set of references, and the availability of their code and data is noteworthy.

I have no recommendations that would require additional data collection, and what I hope are minor requests for additional analysis or clarification of the text. The area that needs the most improvement, at least from my point of view, is the interpretive methods the authors use to present the otherwise useful data—the “Distinct Authors” field in Table 1, for example, and the “Total Preprint to Postprint Ratio” field in Table 2. These are helpful metrics to have, but are presented in unintuitive ways. I elaborate on these points below, but I am nevertheless happy to recommend that this paper be accepted after minor revisions.

Essential revisions:

Lines 14–15: It’s a very interesting idea to use the lack of a large connected coauthorship network to indicate that there’s room for these services to grow, but it’s unclear why this would be true. Does this rely on the implication that large coauthorship networks exist in the published literature but are not yet present on these preprint servers? The authors should clarify (not necessarily in the abstract) their reasoning on this point.

Line 29/71: The authors assert that “COS hosts 25 unique preprint services,” but I can (in early May) only find 23 services on the OSF Preprints homepage, which would make 24 services including OSF Preprints itself. It’s possible I’m not looking in the right place, but the authors should check this figure one more time.

Lines 73–76: The authors lay out reasonable criteria for selecting which preprint services to use, but as far as I can tell, all of these criteria are satisfied by OSF Preprints, the largest service hosted by COS. The authors should clarify why this one was omitted.

Line 81: The definition of “preprint” here—a paper uploaded “prior to being submitted to a journal”—does not match the description on lines 94–97, which relies on “journal publication date.” This would benefit from more precise language.

Line 87: Similarly, this definition of “postprint” incorporates the date a paper was “accepted”, which does not agree with the criteria on lines 96–98, which uses only “journal publication date.”

Line 107(?): The line numbers aren’t applied to the paragraph in between lines 106 and 107 on page 13, but there is a problematic statement toward the end: It says “We then apply the Szymkiewicz-Simpson overlap coefficient to account for disparity in number of papers.” This overlap coefficient does not account for disparity in number of papers—it accounts for disparity in number of categories. Unless I'm mistaken, this should be corrected.

Table 1: “Distinct Authors” is a valuable field to have in this table, but it doesn’t seem helpful to express this number as a percentage—including the total number of distinct authors instead would enable readers to see at a glance how many actual researchers have used each service.

Lines 122–125: Related to the note above, “Unique author percentage” is a confusing metric as presented, and it’s not clear what the reader is meant to get from it. A low “unique author percentage” means more authors have submitted multiple manuscripts, right? The authors should either change these sentences to refer to something like “papers per author” or clarify what we should infer from “unique author percentage.”

Section 3.3: This section may be the most interesting of the manuscript, but it is underdeveloped. The only metric in the text itself is the percentage of papers that appear in “unique venues,” which, like “unique author percentage,” is unclear both in its definition and its utility. It appears the authors could use their existing data to answer far more pressing questions for each service than “unique journals.” Most importantly, this analysis should differentiate between preprints and postprints; the conflation of the two here means we can infer very little about the character of the preprints on each server:

---What percentage of preprints that were published appeared in open-access journals? How does this compare to the percentage of postprints that were published in open-access journals? I would imagine postprints are disproportionately published in closed-access publications, but this effect is lost in the current version of Table 3, which does not differentiate between the two and likely under-represents the proportion of preprint publications that appear in gold-OA publications.

---What percentage of preprints have been published at all? The authors wisely note that publication counts are dependent on authors self-reporting this, but it is an important question for which the data has already been collected. This is currently confounded by the varying number of postprints for which publication is a foregone conclusion. For example, PsyArXiv has lots of published papers, but it appears almost two-thirds of them were posted after publication, which means fewer than 8 percent of PsyArXiv preprints have been published. Even if preprints from, say, the most recent 12 months are excluded, this number is shockingly small. This is almost certainly a metadata problem (i.e. most of those were published, but not updated on the preprint service), but it would be helpful to know what proportion of the preprints is being discussed when publication metrics are described.

Table 5: This table should have a column for “annotations per paper” that divides the number of total annotations by the number of total papers for each service. PsyArXiv is described as “a notable exception,” but when correcting for number of papers on each service, PaleorXiv appears to have an annotation rate about six times higher than PsyArXiv’s.

Line 207–208: The language in the sentence about “COS articles that appear in Gold OA venues” should be clarified—this is the percentage of published COS articles, no? In addition, this measurement should omit postprints from the calculation—it’s hard to see why someone posting in a gold-OA journal would also be compelled to go the “green” OA route for their postprint.

Minor points:

These are notes that I believe would improve the manuscript but that I do not consider necessary for me to be comfortable with its publication.

Line 1: “cyberinfrastructure” seems to be an exaggeration of the technical breadth of what is being described here. Perhaps “web services”?

Lines 7: The phrase “time evolution” here seems to be a misnomer. It’s my understanding that this term refers to changes determined by the structure and internal dynamics of a system that manifest over time, as opposed to the situation being discussed in this paper, which is a static system that has changing inputs. I may be overthinking it, but this sentence seems to describe what could simply be called a “trend.”

Line 11: The authors write, “it is likely that the COS services function as open repositories for peer-reviewed papers that would otherwise be behind a paywall.” The word “likely” here is unnecessary—section 3.3 shows that a small percentage of published papers end up in DOAJ journals, which means the remaining published papers are free copies of peer-reviewed papers that are otherwise behind a paywall. The authors could refer to this finding more definitively here.

Line 33: Numerous “open science” terms are used throughout the manuscript and are either not defined or defined much later in the text. The phrase “green open access” here is one such example. It would be helpful to contrast it to alternatives like “gold open access,” for readers who are not as familiar with the terminology. The same recommendation applies to the use of “hybrid journals” on line 104.

Line 34: I’m not sure it’s accurate to characterize the founding of all the preprint services as happening via “grassroots community action.” MetaArXiv, for example, is maintained by BITSS, which has received millions of dollars in funding. The same could be said for FocUS Archive Preprints (Focused Ultrasound Foundation),  MindRxiv (Mind & Life Institute), and so on.

Figure 2: The very limited data resolution for this figure is surprising—month-level data is used for Figure 1, but this figure only has annual data, which means most services just have a straight line between two points. My suggestion is to combine figures 1 and 2—this could be accomplished by re-creating Figure 1, except instead of each facet containing a single line (the cumulative total of all papers), each facet could have two lines: one for cumulative preprints, and another for cumulative postprints. Readers could then infer this ratio from the difference between the two lines in each facet. This would also serve to clarify other dynamics that are not as obvious from the current version of Figure 2: For example, the current presentation mostly obscures the fact that PaleoRxiv is mostly postprints, which is surprising. I struggled with how best to describe this recommendation: I do not believe it is essential to the publication of this manuscript, but I do think addressing the dramatic difference in resolution between figures 1 and 2 would make this section stronger

Section 3.2: I understand why the preprint/postprint ratio is a helpful metric, but the authors might consider using percentages instead. For example, saying EngrXiv has a 4.1 preprint/postprint ratio requires the reader to do some mental arithmetic to figure out what it means to them—the same information could be expressed by saying, “81 percent of papers uploaded to EngrXiv are preprints.”

Table 3: The table heading “% of papers published in DOAJ journals” is ambiguous about whether it refers to a percentage of all papers, or the percentage of published papers.

Lines 195–197: The tone of these lines is different from the rest of this section—it seems out of place to say “the hope is that the level of adoption by each community continues to grow.” I obviously defer to the discretion of the editor on this matter, but this, and the uncited recommendations to emphasize community outreach, sounds more like advocacy in an otherwise straightforward presentation of the facts.

Lines 242 and 244: The use of the term “connected component,” while it may be technically accurate, is confusing in several places, particularly here. The phrase “many connected components” in this context functionally means “many unconnected components.” It would be much more clear to use another term, but network analysis is not my specialty and I would defer to the authors’ judgement on how best to express this.

Lines 244–246: The authors provide an interesting citation for the assertion that “structural holes” may generate more unique ideas, but there is no citation for the “On the other hand...” statement. If disconnected networks are preferable, why should we care that “certain sub-communities ... have not been adequately reached”? The authors conflate exposure to diverse coauthors with exposure to diverse ideas, and I’m not sure that’s an association that’s supported here. The characterization of these coauthorship networks is done well, but it is a much different task to say what those networks should look like.

Line 210–211: The authors should add a citation to this statement or clarify that it is speculation—i.e. that this data suggests people who post preprints may be more likely to favor open-access journals.

Line 212–213: Related to the previous note. Given the earlier statements that usage of OSF-hosted services is growing to favor preprints, it’s confusing here to see it suggested that the low percentages of gold OA papers means “COS services are a Green OA venue, a place where authors can self archive postprints.” This conclusion may change when these ratios are re-calculated to exclude postprints, but it seems equally reasonable to infer that the low percentages of Gold OA papers simply means that people posting preprints just don’t prioritize open-access publishing.

Author Response

Thank you for the time and effort you put into reviewing our manuscript. We found your comments valuable in revising our submission. Please find below your comments followed in-line by our responses.

Reviewer 1

The area that needs the most improvement, at least from my point of view, is the interpretive methods the authors use to present the otherwise useful data—the “Distinct Authors” field in Table 1, for example, and the “Total Preprint to Postprint Ratio” field in Table 2. These are helpful metrics to have, but are presented in unintuitive ways. I elaborate on these points below, but I am nevertheless happy to recommend that this paper be accepted after minor revisions.

Essential revisions:

Lines 14–15: It’s a very interesting idea to use the lack of a large connected coauthorship network to indicate that there’s room for these services to grow, but it’s unclear why this would be true. Does this rely on the implication that large coauthorship networks exist in the published literature but are not yet present on these preprint servers? The authors should clarify (not necessarily in the abstract) their reasoning on this point.

Network theory - and analysis of real-world networks - informs us that networks         proceed through phases of development and a critical point exists at which     

        networks become robust to failure. Connected component analysis is a means

        of identifying where a network is in its development and placing the COS

        networks in context with other co-authorship and science networks. While

        authors per paper may give similar results, connected components is a

        common technique in network analysis and allows for a more direct

        comparison with other networks.

Line 29/71: The authors assert that “COS hosts 25 unique preprint services,” but I can (in early May) only find 23 services on the OSF Preprints homepage, which would make 24 services including OSF Preprints itself. It’s possible I’m not looking in the right place, but the authors should check this figure one more time.

Thank you for catching our typo. It is 23 unique preprint services (not

counting OSF Preprints). This has been corrected in the text.

Lines 73–76: The authors lay out reasonable criteria for selecting which preprint services to use, but as far as I can tell, all of these criteria are satisfied by OSF Preprints, the largest service hosted by COS. The authors should clarify why this one was omitted.

Yes, the three criteria are satisfied by OSF Preprints. However, our focus is on

        comparison between communities/domains making OSF Preprints and Thesis

        Commons out of scope. We believe OSF Preprints in particular is an

        interesting case study; however, we feel it also raises a number of additional

        questions that would muddy the paper. We omit OSF Preprints to focus on how

        different domains/communities are using preprints and defer OSF Preprints to

        another study. The text has been updated with a fourth criteria detailing this

        thinking for the reader.

Line 81: The definition of “preprint” here—a paper uploaded “prior to being submitted to a journal”—does not match the description on lines 94–97, which relies on “journal publication date.” This would benefit from more precise language.

Thank you for pointing this out. We agree this section needs more precise

        language. We conflated the definitions of preprint and postprint with the steps

        used to classify manuscripts as either preprint or postprint. We have clarified

        the language in this section. We use the Tennant et al. definition of

        preprint/postprint and use journal publication date to classify a manuscript. If

        no journal publication date exists then the manuscript is classified as a

        preprint.

Line 87: Similarly, this definition of “postprint” incorporates the date a paper was “accepted”, which does not agree with the criteria on lines 96–98, which uses only “journal publication date.”

This is imprecise wording on our part. Our analysis is based off of the

        publication date, not the date of acceptance. The text has been updated to

        clarify this.

Line 107(?): The line numbers aren’t applied to the paragraph in between lines 106 and 107 on page 13, but there is a problematic statement toward the end: It says “We then apply the Szymkiewicz-Simpson overlap coefficient to account for disparity in number of papers.” This overlap coefficient does not account for disparity in number of papers—it accounts for disparity in number of categories. Unless I'm mistaken, this should be corrected.

You are correct. Thank you for catching this. We are actually accounting for the

        disparity in keywords (Discipline tags) not the disparity in number of papers.

        The text has been updated to fix this.

Table 1: “Distinct Authors” is a valuable field to have in this table, but it doesn’t seem helpful to express this number as a percentage—including the total number of distinct authors instead would enable readers to see at a glance how many actual researchers have used each service.

The table has been updated to reflect the total number of distinct authors.

Lines 122–125: Related to the note above, “Unique author percentage” is a confusing metric as presented, and it’s not clear what the reader is meant to get from it. A low “unique author percentage” means more authors have submitted multiple manuscripts, right? The authors should either change these sentences to refer to something like “papers per author” or clarify what we should infer from “unique author percentage.”

We changed “unique authors” to “distinct authors” to be consistent with the

        wording used in Table 1. We view distinct author percentages as a proxy for

        service uptake and have updated the text to clarify what readers should infer

        from these numbers.

Section 3.3: This section may be the most interesting of the manuscript, but it is underdeveloped. The only metric in the text itself is the percentage of papers that appear in “unique venues,” which, like “unique author percentage,” is unclear both in its definition and its utility. It appears the authors could use their existing data to answer far more pressing questions for each service than “unique journals.” Most importantly, this analysis should differentiate between preprints and postprints; the conflation of the two here means we can infer very little about the character of the preprints on each server:

---What percentage of preprints that were published appeared in open-access journals? How does this compare to the percentage of postprints that were published in open-access journals? I would imagine postprints are disproportionately published in closed-access publications, but this effect is lost in the current version of Table 3, which does not differentiate between the two and likely under-represents the proportion of preprint publications that appear in gold-OA publications.

---What percentage of preprints have been published at all? The authors wisely note that publication counts are dependent on authors self-reporting this, but it is an important question for which the data has already been collected. This is currently confounded by the varying number of postprints for which publication is a foregone conclusion. For example, PsyArXiv has lots of published papers, but it appears almost two-thirds of them were posted after publication, which means fewer than 8 percent of PsyArXiv preprints have been published. Even if preprints from, say, the most recent 12 months are excluded, this number is shockingly small. This is almost certainly a metadata problem (i.e. most of those were published, but not updated on the preprint service), but it would be helpful to know what proportion of the preprints is being discussed when publication metrics are described.

Thank you for the suggestion. We have modified our analysis in section 3.3 to

        look at the preprint vs. postprint OA questions presented by the reviewer.

Table 5: This table should have a column for “annotations per paper” that divides the number of total annotations by the number of total papers for each service. PsyArXiv is described as “a notable exception,” but when correcting for number of papers on each service, PaleorXiv appears to have an annotation rate about six times higher than PsyArXiv’s.

Thank you for the suggestion. We have added an “annotations per paper”

        column to Table 5. When viewed in this context, we agree that PaleorXiv is

        notable. We have updated the text to reflect this.

Line 207–208: The language in the sentence about “COS articles that appear in Gold OA venues” should be clarified—this is the percentage of published COS articles, no? In addition, this measurement should omit postprints from the calculation—it’s hard to see why someone posting in a gold-OA journal would also be compelled to go the “green” OA route for their postprint.

We agree that it’s hard to see why someone posting in a gold-OA journal would

        also be compelled to go Green OA as well. Yet, this is answerable with our

        data so we investigate this in our revised Table 3.

Minor points:

These are notes that I believe would improve the manuscript but that I do not consider necessary for me to be comfortable with its publication.

Line 1: “cyberinfrastructure” seems to be an exaggeration of the technical breadth of what is being described here. Perhaps “web services”?

Changed to “web-based systems”

Lines 7: The phrase “time evolution” here seems to be a misnomer. It’s my understanding that this term refers to changes determined by the structure and internal dynamics of a system that manifest over time, as opposed to the situation being discussed in this paper, which is a static system that has changing inputs. I may be overthinking it, but this sentence seems to describe what could simply be called a “trend.”

Changed to “trend”

Line 11: The authors write, “it is likely that the COS services function as open repositories for peer-reviewed papers that would otherwise be behind a paywall.” The word “likely” here is unnecessary—section 3.3 shows that a small percentage of published papers end up in DOAJ journals, which means the remaining published papers are free copies of peer-reviewed papers that are otherwise behind a paywall. The authors could refer to this finding more definitively here.

The word “likely” has been removed.

Line 33: Numerous “open science” terms are used throughout the manuscript and are either not defined or defined much later in the text. The phrase “green open access” here is one such example. It would be helpful to contrast it to alternatives like “gold open access,” for readers who are not as familiar with the terminology. The same recommendation applies to the use of “hybrid journals” on line 104.

We have added three bullet points to the Methods section defining the open

        science terms used. We have also included a reference to where we draw our

        definitions of these terms.

Line 34: I’m not sure it’s accurate to characterize the founding of all the preprint services as happening via “grassroots community action.” MetaArXiv, for example, is maintained by BITSS, which has received millions of dollars in funding. The same could be said for FocUS Archive Preprints (Focused Ultrasound Foundation),  MindRxiv (Mind & Life Institute), and so on.

Changed to “many” services were started via “grassroots community action”

Figure 2: The very limited data resolution for this figure is surprising—month-level data is used for Figure 1, but this figure only has annual data, which means most services just have a straight line between two points. My suggestion is to combine figures 1 and 2—this could be accomplished by re-creating Figure 1, except instead of each facet containing a single line (the cumulative total of all papers), each facet could have two lines: one for cumulative preprints, and another for cumulative postprints. Readers could then infer this ratio from the difference between the two lines in each facet. This would also serve to clarify other dynamics that are not as obvious from the current version of Figure 2: For example, the current presentation mostly obscures the fact that PaleoRxiv is mostly postprints, which is surprising. I struggled with how best to describe this recommendation: I do not believe it is essential to the publication of this manuscript, but I do think addressing the dramatic difference in resolution between figures 1 and 2 would make this section stronger

Most of the services in our study are only two years old. Moreover, several of

        these services were front-loaded with postprints so there was content available

        when the services launched. Given the short time frames and the front-loading,         we feel monthly resolution in figure 2 would lead to a difficult to interpret - and         potentially misleading - figure. We believe that annual resolution averages out         these effects. We understand the reviewers concern; however, we would like to         keep figure 2 unchanged.

Section 3.2: I understand why the preprint/postprint ratio is a helpful metric, but the authors might consider using percentages instead. For example, saying EngrXiv has a 4.1 preprint/postprint ratio requires the reader to do some mental arithmetic to figure out what it means to them—the same information could be expressed by saying, “81 percent of papers uploaded to EngrXiv are preprints.”

Thank you for the suggestion. We see how percent could be more informative

        for some readers; however, we would prefer to keep the ratio language.

Table 3: The table heading “% of papers published in DOAJ journals” is ambiguous about whether it refers to a percentage of all papers, or the percentage of published papers.

Table headings have been changed to clarify.

Lines 195–197: The tone of these lines is different from the rest of this section—it seems out of place to say “the hope is that the level of adoption by each community continues to grow.” I obviously defer to the discretion of the editor on this matter, but this, and the uncited recommendations to emphasize community outreach, sounds more like advocacy in an otherwise straightforward presentation of the facts.

The wording has been changed so that the tone remains the same throughout

        the paper.

Lines 242 and 244: The use of the term “connected component,” while it may be technically accurate, is confusing in several places, particularly here. The phrase “many connected components” in this context functionally means “many unconnected components.” It would be much more clear to use another term, but network analysis is not my specialty and I would defer to the authors’ judgement on how best to express this.

We see the reviewer’s point. However, we prefer to use the formal

        mathematical and accepted network science term here.

Lines 244–246: The authors provide an interesting citation for the assertion that “structural holes” may generate more unique ideas, but there is no citation for the “On the other hand...” statement. If disconnected networks are preferable, why should we care that “certain sub-communities ... have not been adequately reached”? The authors conflate exposure to diverse coauthors with exposure to diverse ideas, and I’m not sure that’s an association that’s supported here. The characterization of these coauthorship networks is done well, but it is a much different task to say what those networks should look like.

This section has been revised to provide additional background on why

        connected components are important and what the reader can take away from

        the analysis.

Line 210–211: The authors should add a citation to this statement or clarify that it is speculation—i.e. that this data suggests people who post preprints may be more likely to favor open-access journals.

This has been revised based on additional analysis of preprint vs. postprint

        DOAJ publishing.

Line 212–213: Related to the previous note. Given the earlier statements that usage of OSF-hosted services is growing to favor preprints, it’s confusing here to see it suggested that the low percentages of gold OA papers means “COS services are a Green OA venue, a place where authors can self archive postprints.” This conclusion may change when these ratios are re-calculated to exclude postprints, but it seems equally reasonable to infer that the low percentages of Gold OA papers simply means that people posting preprints just don’t prioritize open-access publishing.

This has been revised based on additional analysis of preprint vs. postprint

        DOAJ publishing.

Reviewer 2 Report

I noticed a couple typos:

Line 16: "each manuscripts"

Line 127: "us calculate"

Line 139: "published documents in"

Line 146: "these ratio"

Line 198: "indicate"

Line 217: "version so of"

Line 226: "The update" (updating?)

Line 271: "reprint"

In addition, sometimes after a period there is 1 space, 2 spaces, or no spaces.  I'm not sure if this resulted from the document conversion.

I wasn't specifically looking for typos so the manuscript should probably be proofread before publishing.

I would like the authors to clarify their definition of postprints, since they define postprints six different ways throughout the article.

1) In lines 5-6 they write: "postprints, non-typeset versions of peer-reviewed manuscripts."

2) In line 33 they write: "postprints, self-archived final versions of papers that are behind paywalls"

3) In lines 86-88 they write: "We define a Postprint as any manuscript uploaded to a COS system after being accepted or published in a peer-reviewed journal."

4) In lines 96-97 they write: "If the COS submission date occurs after the journal publication date, the manuscript is considered a postprint"

5) In lines 214-215 they write: "By updating a preprint to reflect the published version of an article, an author is creating a postprint"

6) In lines 213-214 they write: "postprints (the non-typeset versions of peer-reviewed publications that would otherwise reside behind a paywall)"

The authors state that their definition of postprints is similar to Tennant et al., however only their first definition matches the Tennant et al. definition, which seems to be the generally accepted definition of postprints (an accepted manuscript prior to typesetting).

Definitions 2 and 6 of the authors imply the term "postprints" only applies to manuscripts behind paywalls, which is wrong.  In addition, the term "final version" could use some clarification.  The content of a postprint and published manuscript is the same, yet I think most people would consider "final version" to refer to the published manuscript.

Definitions 3 and 4 are in conflict with each other.  The date of acceptance is not the date of publication.  There is usually a delay between these dates, sometimes a significant one.  I haven't used the Unpaywall API, but I assume the authors changed their definition of postprints mid-manuscript because the API only provided the publication date, not the accepted date.

I encountered the same issue when I looked for postprints posted at bioRxiv with the CrossRef API: http://www.prepubmed.org/shame/
To find the date of acceptance I had to manually go to the publisher website, which I wouldn't recommend since that is time consuming.  One thing the authors could do however is show a graph of posting date to publication date.  If there are a lot of of manuscripts posted near the publication date I would be concerned these are mostly postprints, and it would be interesting to see if this differs among the servers.

In addition to underestimating the number of postprints because of the delay between acceptance and publication, as the authors note the COS API may not include a publication DOI even if the manuscript has been published.  This is also a problem with bioRxiv, and the authors here found that upwards of 53% of unlinked manuscripts were not linked to their published version (https://elifesciences.org/articles/45133).  I'd like the authors to perform a similar survey to get a sense of how often the COS API is not linking to the published version.  If high enough, the authors' estimations of percentages of postprints could be grossly underestimated.  Although it is tempting to tell a story of increasing preprint uptake in these disciplines, we should do our best to see if this is actually the case.

I find Figure 2 to be misleading.  The figure purports to show that the preprint to postprint ratio is increasing, but presumably it takes time for the COS API to include the published DOI, and with the authors' methodology every single manuscript missing a published DOI gets labeled a preprint.  Because manuscripts posted more recently are possibly more likely to be unlinked to their published version, and because all of these then get labeled preprints, I think the trend seen in Figure 2 can be partially or wholly explained as an artifact of the authors' methods.

Given the difficulty of accurately labeling manuscripts postprints, I think it is fair for the authors to label manuscripts posted after publication as postprints, but instead of changing the definition of postprints they should state that their analysis likely underestimates how many postprints there are since the accepted date could have been months or even years before the publication date, and potentially a lot of manuscripts are not linked to their published versions, and this underestimation could vary depending on the server.

Given my experience with bioRxiv preprints, my guess would be a lot of authors are posting preprints as soon as the manuscript is accepted, and thus are posting a postprint and not a preprint.  The question is why would authors bother to post their manuscript to a server if they know it's about to be published?  The reasons for this are likely selfish.  By only posting after acceptance they don't have to worry about "scooping" (although a preprint protects against scooping).  And posting to a preprint server allows them to get a wave of attention, potentially even media attention.  The posting also virtue signals to the open science community.

When the discussion of postprints comes up people generally assume authors are doing it to provide the public with a freely available version of their work.  My view of scientists is much more cynical, but it would be interesting to see if the general assumption is correct.  I'd like to see the authors look at how many of their manuscripts labeled postprints were published in closed journals, and whether this ratio is different from preprints.  If most postprints are published in closed journals it would indicate that the postprints exist to provide an open-access version.  If there is no difference between postprints and preprints I would say that suggests that authors are posting postprints for other (likely selfish) reasons.

The fifth definition of postprints by the authors is interesting.  I'm not comfortable with the idea that a preprint can become a postprint.  I would say the most recent version of the manuscript posted to the server is a postprint.

#######

When describing how the authors decided which COS preprint systems to analyze the authors state: "Third, the manuscripts must be accessible through the OSF Application Programming Interface".  This implies that some of the 25 COS preprint systems cannot be accessed via the API.  Which ones are these?  And why can't they be accessed? Is the COS aware of this?

The authors write: "Errors in the journal publication dates reported by the Unpaywall API are reported here as unclassified".  How do the authors know the dates are in error?  What types of errors are these?  Can the dates in the Unpaywall API be crosschecked with the CrossRef API?  Here is an example of a CrossRef call: https://api.crossref.org/v1/works/10.1101/096818

#######

I'm not sure if Table 4 is providing useful information or not.  For example, even before seeing the Table I could have told you PsyArXiv would have the largest node because of preprints like these:
https://psyarxiv.com/mky9j
https://psyarxiv.com/9s3y6/
https://psyarxiv.com/9654g/
https://psyarxiv.com/q4emc/

My point is I'm not sure looking at the largest node tells you very much, although I haven't done any network analysis.  I suspect that just looking at number of authors per paper would give the same result.  Maybe eliminating papers with an outlier number of authors would be more informative?

Regarding the network analysis, I'm curious if the authors had any difficulty collapsing author names.  For example, sometimes a person will provide their middle initial, or sometimes they won't.  Maybe someone will write their name as Nick, and sometimes Nicholas.  I assume the posting author must have an OSF account, so presumably every time they post a manuscript their name doesn't change.  However, it is possible someone has created multiple OSF accounts.  It seems that coauthors do not need to create accounts, so I could see these names changing between manuscripts.

If the authors did not have any trouble collapsing names I'd like them to explain why they were so successful.

######

Some other analyses which could be interesting but may be outside the scope of this paper:

-Are there differences between preprints, published preprints, and postprints.  Length, number of authors? Impact fact of journal (for published preprints and postprints).

-Number of different downloads for the different servers?  Altmetrics of preprints in the different servers?
The authors note that download statistics are not available through the API, but surely they can be scraped from the preprint pages?  If the authors do look at downloads they should investigate whether downloads can be manipulated, as was done here: https://liorpachter.wordpress.com/2019/01/29/technologies-of-narcissism/

-Does number of comments correlate with download numbers?  I.e. it is likely that only highly read manuscripts get commented on.

Author Response

Thank you for your thorough review. We found your suggestions very informative in guiding our revisions. Please find below your comments with our responses in-line.

Reviewer 2

I noticed a couple typos:

Line 16: "each manuscripts"

Line 127: "us calculate"

Line 139: "published documents in"

Line 146: "these ratio"

Line 198: "indicate"

Line 217: "version so of"

Line 226: "The update" (updating?)

Line 271: "reprint"

These typos have been corrected in the text.

I would like the authors to clarify their definition of postprints, since they define postprints six different ways throughout the article.

1) In lines 5-6 they write: "postprints, non-typeset versions of peer-reviewed manuscripts."

2) In line 33 they write: "postprints, self-archived final versions of papers that are behind paywalls"

3) In lines 86-88 they write: "We define a Postprint as any manuscript uploaded to a COS system after being accepted or published in a peer-reviewed journal."

4) In lines 96-97 they write: "If the COS submission date occurs after the journal publication date, the manuscript is considered a postprint"

6) In lines 213-214 they write: "postprints (the non-typeset versions of peer-reviewed publications that would otherwise reside behind a paywall)"

Definitions 2 and 6 of the authors imply the term "postprints" only applies to manuscripts behind paywalls, which is wrong.  In addition, the term "final version" could use some clarification.  The content of a postprint and published manuscript is the same, yet I think most people would consider "final version" to refer to the published manuscript.

Definitions 3 and 4 are in conflict with each other.  The date of acceptance is not the date of publication.  There is usually a delay between these dates, sometimes a significant one.  I haven't used the Unpaywall API, but I assume the authors changed their definition of postprints mid-manuscript because the API only provided the publication date, not the accepted date.

The authors state that their definition of postprints is similar to Tennant et al., however only their first definition matches the Tennant et al. definition, which seems to be the generally accepted definition of postprints (an accepted manuscript prior to typesetting).

Thank you for pointing out this inconsistency. The text has been updated to

        reflect the Tennant et al. definition of postprint. We define a postprint, similar to

        Tennant et al., as a non-typeset version of a peer-reviewed manuscript. We             now define a method to determine if a manuscript meets this definition.

        Specifically, our method is to compare the COS submission date to the peer-

        reviewed publication date. If the COS date is earlier then the manuscript is a

        preprint. If no peer-reviewed publication date exists, we also classify the

        manuscript as a preprint. We incorrectly worded the steps in this method as

        definitions for a postprint. It is more precise to call these classifying steps

        towards the Tennant et al. definition. The text has been updated to clarify this

        point.

I encountered the same issue when I looked for postprints posted at bioRxiv with the CrossRef API: http://www.prepubmed.org/shame/

To find the date of acceptance I had to manually go to the publisher website, which I wouldn't recommend since that is time consuming.  One thing the authors could do however is show a graph of posting date to publication date.  If there are a lot of of manuscripts posted near the publication date I would be concerned these are mostly postprints, and it would be interesting to see if this differs among the servers.

In addition to underestimating the number of postprints because of the delay between acceptance and publication, as the authors note the COS API may not include a publication DOI even if the manuscript has been published.  This is also a problem with bioRxiv, and the authors here found that upwards of 53% of unlinked manuscripts were not linked to their published version (https://elifesciences.org/articles/45133).  I'd like the authors to perform a similar survey to get a sense of how often the COS API is not linking to the published version.  If high enough, the authors' estimations of percentages of postprints could be grossly underestimated.  Although it is tempting to tell a story of increasing preprint uptake in these disciplines, we should do our best to see if this is actually the case.

Thank you for the suggestion. We chose 12 random preprints papers from

         each system - excluding lawarxiv - for a total of 96 papers. We excluded

         lawarxiv as this domain often published in venues not assigning DOIs. The 96

         preprints were manually entered into Google Scholar to identify cases where

         published versions of the paper existed, but the authors did not update the

         corresponding COS record. 35 such cases were found (36%). The text has

         been updated accordingly.

I find Figure 2 to be misleading.  The figure purports to show that the preprint to postprint ratio is increasing, but presumably it takes time for the COS API to include the published DOI, and with the authors' methodology every single manuscript missing a published DOI gets labeled a preprint.  Because manuscripts posted more recently are possibly more likely to be unlinked to their published version, and because all of these then get labeled preprints, I think the trend seen in Figure 2 can be partially or wholly explained as an artifact of the authors' methods.

We are a little unclear as to the reviewers suggestion here. The reviewer is

        correct that every paper missing a publishers DOI is labeled as a preprint.

        COS Authors are responsible for linking the DOI for their published paper to

        their preprint, and this question is asked as the paper is deposited in the COS

        service (i.e., if the paper is a postprint). Once the authors change this

        information, it is then obtainable through the COS API. As explained above, we

        do check the compliance of authors with a random sampling of 12 preprints per

        service and manually investigate how many have peer-reviewed version

        despite the authors not linking to those versions. We now include this number,

        36%, in the text to give an estimate of how much we likely underestimate.

        We determine the publication date of all papers labelled ‘postprint’ by querying

        the Unpaywall DOI using the peer-reviewed DOI. So we only assign the peer-

        reviewed publication date to a manuscript that has a peer reviewed DOI.

Given the difficulty of accurately labeling manuscripts postprints, I think it is fair for the authors to label manuscripts posted after publication as postprints, but instead of changing the definition of postprints they should state that their analysis likely underestimates how many postprints there are since the accepted date could have been months or even years before the publication date, and potentially a lot of manuscripts are not linked to their published versions, and this underestimation could vary depending on the server.

Given my experience with bioRxiv preprints, my guess would be a lot of authors are posting preprints as soon as the manuscript is accepted, and thus are posting a postprint and not a preprint.  The question is why would authors bother to post their manuscript to a server if they know it's about to be published?  The reasons for this are likely selfish.  By only posting after acceptance they don't have to worry about "scooping" (although a preprint protects against scooping).  And posting to a preprint server allows them to get a wave of attention, potentially even media attention.  The posting also virtue signals to the open science community.

When the discussion of postprints comes up people generally assume authors are doing it to provide the public with a freely available version of their work.  My view of scientists is much more cynical, but it would be interesting to see if the general assumption is correct.  I'd like to see the authors look at how many of their manuscripts labeled postprints were published in closed journals, and whether this ratio is different from preprints.  If most postprints are published in closed journals it would indicate that the postprints exist to provide an open-access version.  If there is no difference between postprints and preprints I would say that suggests that authors are posting postprints for other (likely selfish) reasons.

Thank you for this suggestion. We have done this analysis and have updated

        the text and tables accordingly.

The fifth definition of postprints by the authors is interesting.  I'm not comfortable with the idea that a preprint can become a postprint.  I would say the most recent version of the manuscript posted to the server is a postprint.

We agree that the wording was confusing. We were conflating multiple

        situations. The text has been updated to clarify.

#######

When describing how the authors decided which COS preprint systems to analyze the authors state: "Third, the manuscripts must be accessible through the OSF Application Programming Interface".  This implies that some of the 25 COS preprint systems cannot be accessed via the API.  Which ones are these?  And why can't they be accessed? Is the COS aware of this?

The OSF API documentation erroneously pointed to an out of date reference

        that did not include Thesis Commons in the list of available services reachable

        by the API. In fact, Thesis Commons was available, but we did not know this

        nor did we know how to query it given the error in the documentation. We

        reached out to COS at the beginning of our study, but they were not able to

        correct the issue until just before our revisions were due. All COS systems are

        now accessible via the API. However, we do not feel that Thesis Commons

        and OSF Preprints are applicable for our present study where our focus is on

        domain/community differences. General purpose services like Thesis

        Commons and OSF Preprints - and why an author would choose them over

        discipline specific services - is deferred for future research.

The authors write: "Errors in the journal publication dates reported by the Unpaywall API are reported here as unclassified".  How do the authors know the dates are in error?  What types of errors are these?  Can the dates in the Unpaywall API be crosschecked with the CrossRef API?  Here is an example of a CrossRef call: https://api.crossref.org/v1/works/10.1101/096818

This is a typo on our part. “Errors in the journal publication dates” should

have read “Errors when the journal publication date was not obtainable by

the Unpaywall API.” We have updated the text to fix this. Specifically, we

encountered instances where the Unpaywall API did not recognize

valid DOIs

#######

I'm not sure if Table 4 is providing useful information or not.  For example, even before seeing the Table I could have told you PsyArXiv would have the largest node because of preprints like these:

https://psyarxiv.com/mky9j

https://psyarxiv.com/9s3y6/

https://psyarxiv.com/9654g/

https://psyarxiv.com/q4emc/

My point is I'm not sure looking at the largest node tells you very much, although I haven't done any network analysis.  I suspect that just looking at number of authors per paper would give the same result.  Maybe eliminating papers with an outlier number of authors would be more informative?

Network theory - and analysis of real-world networks - informs us that

networks proceed through phases of development and a critical point exists

at which networks become robust to failure. Connected component analysis

is a means of identifying where a network is in its development and placing

the COS networks in context with other co-authorship and science

networks. While authors per paper may give similar results, connected

components is a common technique in network analysis and allows for a

more direct comparison with other networks.

Regarding the network analysis, I'm curious if the authors had any difficulty collapsing author names.  For example, sometimes a person will provide their middle initial, or sometimes they won't.  Maybe someone will write their name as Nick, and sometimes Nicholas.  I assume the posting author must have an OSF account, so presumably every time they post a manuscript their name doesn't change.  However, it is possible someone has created multiple OSF accounts.  It seems that coauthors do not need to create accounts, so I could see these names changing between manuscripts.

If the authors did not have any trouble collapsing names I'd like them to explain why they were so successful.

This is an excellent point and one we failed to mention in the text. We did

use a semi-autonomous means of disambiguation and found a small

number of author name conflicts. Specifically, we looked at all pairwise

comparison of author names in our data. Potential matches - based on last

name exact match and similarity of first initial were marked for manual

inspection. Following manual inspection, we had identified 63 conflicting

names, which we could not conclusively disambiguate. This introduces an

uncertainty into our analysis; yet, we feel the small number of conflicting

names gives us a good approximation of the true network. We did,     

however, add text describing our procedure so readers are aware of the

uncertainty in our analysis.

######

Some other analyses which could be interesting but may be outside the scope of this paper:

-Are there differences between preprints, published preprints, and postprints.  Length, number of authors? Impact fact of journal (for published preprints and postprints).

We agree these are interesting questions; however, they are outside the scope

        of this paper. Given the limited time we have for revisions (10 days), we don’t

        feel we can do justice to this analysis. We would like to reserve this for future

        research.

-Number of different downloads for the different servers?  Altmetrics of preprints in the different servers?

The authors note that download statistics are not available through the API, but surely they can be scraped from the preprint pages?  If the authors do look at downloads they should investigate whether downloads can be manipulated, as was done here: https://liorpachter.wordpress.com/2019/01/29/technologies-of-narcissism/

-Does number of comments correlate with download numbers?  I.e. it is likely that only highly read manuscripts get commented on.

Excellent questions. We were thinking along similar lines; however, COS     

        informed us that download counts were only recently added to the API and the

        feature is still experimental. During our testing, we found several

        inconsistencies between download counts reported by the API and download

        counts listed on the websites. We have reported these inconsistencies to COS

        and they are working the issue. We tried resorting to scraping the preprint

        pages; however, the download counts do not appear in the raw HTML. They

        are dynamically generated (we believe through Javascript). As such, we don’t

        feel that these questions can be addressed at this time (at least not without a

        large-scale manual effort) and we are deferring these questions for future

        research.

Round 2

Reviewer 2 Report

With regards to my comment on Figure 2 in my previous comments:

I was suggesting that there may be a lag between an author posting a postprint, and this information being displayed in the API.  I thought maybe an author later realized they should have provided a DOI and updated their manuscript or the COS realized it was a postprint and updated the API themselves.  If the only time postprint authors are adding the DOI is at submission then this concern doesn't apply.

Another thing that is possible is maybe author behavior is not constant.  For example, maybe it is more likely now that authors posting a postprint will not provide a DOI.  This could occur if posting a preprint is now considered a "cool thing to do" and authors don't want people to notice their preprints are actually postprints.  As a result, I am still concerned about whether the trend seen in Figure 2 shows a true increase in preprints vs. postprints.

##

I looked over reviewer 1's comments and noticed he/she thought OSF Preprints met all of the authors' requirements for inclusion in the study.  This is actually false.  OSF Preprints does have manuscripts in Spanish, which would seem to violate the English language requirement of the authors.  And actually some of the other hosted services also contain Spanish language manuscripts.  The authors may want to clarify whether their English requirement means only English manuscripts or mostly English manuscripts.

##

I appreciate the authors checking 96 manuscripts which had been classified preprints and seeing if they have a DOI.  The authors identified 35 which were missing a DOI, but I would like to know if these 35 are preprints or postprints.  I.e. are these cases of preprint authors forgetting to update their preprint or are they postprints where the authors didn't provide a DOI at submission.

Author Response

With regards to my comment on Figure 2 in my previous comments:

I was suggesting that there may be a lag between an author posting a postprint, and this information being displayed in the API.  I thought maybe an author later realized they should have provided a DOI and updated their manuscript or the COS realized it was a postprint and updated the API themselves.  If the only time postprint authors are adding the DOI is at submission then this concern doesn't apply.

Another thing that is possible is maybe author behavior is not constant.  For example, maybe it is more likely now that authors posting a postprint will not provide a DOI.  This could occur if posting a preprint is now considered a "cool thing to do" and authors don't want people to notice their preprints are actually postprints.  As a result, I am still concerned about whether the trend seen in Figure 2 shows a true increase in preprints vs. postprints.

Thank you for clarifying. There are a few discussion points here.

As far as we are aware, any change made by an author is immediately visible in the API. (i.e., if i submit a preprint to a service, a subsequent call to the API will show the preprint)

COS does not update the data records for a document themselves (i.e., for linking preprints to the DOI of a subsequent postprint) . All updates come from authors who can return to their preprint system at any time to add a peer reviewed DOI.

Authors can (and are expected to) return to their data records and update with the peer reviewed DOI.

We agree that authors might intentionally or unintentionally omit the peer-reviewed DOI during the initial submission process of a postprint (though the COS interface is not onerous — asking for peer reviewed DOI is one of only a handful of Qs that are asked). We tend to think that errors in reporting peer-reviewed DOIs are likely errors of omission vs errors of malice — we have no indication (through interviews, surveys, hearsay) to suggest to readers that any foul play is going on. We now add a line in the manuscript expressing this:

“Preprint and postprint classifications are based off of data available via the OSF API. In other words, we are dependent on authors either 1) voluntarily returning to their COS system and updating their records with the peer-reviewed DOI (for a preprint) or 2) providing the peer-reviewed DOI for their postprint submission. Some authors may not comply with this protocol — either intentionally or unintentionally — for a variety of reasons (e.g., they may forget, or the final  publication venues may not mint a DOI). For this reason our results likely underestimate the total number of COS manuscripts that appear in peer-reviewed journals. “

Further analysis of the 96 manuscripts we manually checked - see third comment - reveals that postprints were incorrectly classified as preprints in 10 out of 35 cases. We acknowledge the concern with Figure 2 and we recognize there is uncertainty with the preprint vs. postprint trend. This new manually checked data suggests that the fig 2 data points towards it being a true trend. Regardless, we think the concerns of the reviewer are valid and we now mention this is in the text :

“These preprint to postprint ratios are not fixed through time. Narock et al (2019) discussed how the initial growth of EarthArXiv was driven by a frontloading of the service with postprints for the launch of the service. As a result it took time for preprints to start outpacing postprint uploads on EarthArXiv. We investigate the temporal evolution of these ratios in Figure 2. First we examine the evolution of the cumulative ratio (Figure 2a). Three services have trends of increasingly being used as preprint servers (PsyArXiv, EarthArXiv, LISSA). Slightly downward trends are seen for the remaining five services. Submission ratios are also examined yearly, which might be useful if submission trends are variable (i.e., frontloading a service with postprints). Again, three services show trends of being used more for preprints — PsyArXiv, EarthArXiv, LISSA — and the growth in preprint usage is more pronounced (Figure 2b). We want to remind readers that postprint designation requires the peer-reviewed DOI to be entered by the author in the COS service, so these results may overestimate the number of preprints in each service.

##

I looked over reviewer 1's comments and noticed he/she thought OSF Preprints met all of the authors' requirements for inclusion in the study.  This is actually false.  OSF Preprints does have manuscripts in Spanish, which would seem to violate the English language requirement of the authors.  And actually some of the other hosted services also contain Spanish language manuscripts.  The authors may want to clarify whether their English requirement means only English manuscripts or mostly English manuscripts.

Thank you for pointing this out. We will add a clarifying sentence to the text.

##

I appreciate the authors checking 96 manuscripts which had been classified preprints and seeing if they have a DOI.  The authors identified 35 which were missing a DOI, but I would like to know if these 35 are preprints or postprints.  I.e. are these cases of preprint authors forgetting to update their preprint or are they postprints where the authors didn't provide a DOI at submission.

25 were preprints - the manuscript was submitted to a COS system before it

        appeared in a journal. In these 25 cases, the authors failed to return to their

        COS system to add the peer reviewed DOI. 10 case were postprints - the

        authors uploaded an already peer reviewed article to a COS system without     

        supplying the peer reviewed DOI. We will add a sentence to the text clarifying

        this for the reader.